# Klotho Protein Decreases MMP-Mediated Degradation of Contractile Proteins during Ischaemia/Reperfusion Injury to the Cardiomyocytes

**DOI:** 10.3390/ijms232415450

**Published:** 2022-12-07

**Authors:** Agnieszka Olejnik, Anna Krzywonos-Zawadzka, Marta Banaszkiewicz, Iwona Bil-Lula

**Affiliations:** Division of Clinical Chemistry and Laboratory Haematology, Department of Medical Laboratory Diagnostics, Faculty of Pharmacy, Wroclaw Medical University, 50-556 Wroclaw, Poland

**Keywords:** Klotho protein, cardioprotection, ischaemia/reperfusion injury, matrix metalloproteinases, heart conctractile proteins

## Abstract

Ischaemia, followed by reperfusion, causes the generation of reactive oxygen species, overproduction of peroxynitrite, activation of matrix metalloproteinases (MMPs), and subsequently the degradation of heart contractile proteins in the cardiomyocytes. Klotho is a membrane-bound or soluble protein that regulates mineral metabolism and has antioxidative activity. This study aimed to examine the influence of Klotho protein on the MMP-mediated degradation of contractile proteins during ischaemia/reperfusion injury (IRI) to the cardiomyocytes. Human cardiac myocytes (HCM) underwent in vitro chemical IRI (with sodium cyanide and deoxyglucose), with or without the administration of recombinant Klotho protein. The expression of MMP genes, the expression and activity of MMP proteins, as well as the level of contractile proteins such as myosin light chain 1 (MLC1) and troponin I (TnI) in HCM were measured. Administration of Klotho protein resulted in a decreased activity of MMP-2 and reduced the release of MLC1 and TnI that followed in cells subjected to IRI. Thus, Klotho protein contributes to the inhibition of MMP-dependent degradation of contractile proteins and prevents injury to the cardiomyocytes during IRI.

## 1. Introduction

An injury to the myocardium during ischaemia/reperfusion involves morphological, metabolic and contractile disorders [1]. An important factor contributing to ischaemia/reperfusion injury (IRI) is an excessive formation of reactive oxygen species (ROS) and oxidative stress. ROS can directly modify heart contractile proteins, activate kinases and transcription factors of cardiac hypertrophy, and subsequently induce apoptosis [2,3]. The progression of myocardial failure and remodelling is associated with the proliferation of cardiac fibroblasts and activation of matrix metalloproteinases (MMPs) due to oxidative stress [3]. It has been shown that the overproduction of both ROS and reactive nitrogen species (RNS) leads to the formation of peroxynitrite (ONOO)^−^, a powerful oxidant and nitrating agent [3]. ONOO^−^ mediates IRI, especially through the activation of MMPs [4,5]. MMPs, in common with MMP-2 and MMP-9, are proteolytic enzymes that degrade extracellular proteins and play an important role in the remodelling of the extracellular matrix [6]. Nevertheless, oxidative/nitrosative stress activates MMPs and mediates the proteolysis of contractile proteins in the cardiomyocytes. It is known that the degradation of proteins such as troponin, titin or myosin light chains (MLCs) by proteolytic enzymes is a major contributor to contractile dysfunction during an IRI to the heart [3,4,5,7].

Klotho is a membrane-bound or soluble anti-aging protein. Membrane Klotho serves as a coreceptor for fibroblast growth factor 23 (FGF23) and maintains ion homeostasis. Klotho and FGF23 induce a negative phosphate and calcium balance [8,9,10]. Moreover, soluble Klotho regulates mineral metabolism independently of FGF23. It can catalyze the hydrolysis of sialic acid or β-D-glucuronic acid residues from the glycan chains in polysaccharides. Thus, Klotho can modify the renal and intestinal ion transporters [11]. Recent studies showed a decrease in ROS generation, and reduced oxidative stress and apoptosis after Klotho administration in several cell lines and animal models [12,13,14,15]. We have previously reported enhanced expression of Klotho in human cardiomyocytes subjected to IRI [16]. We proposed that the increased, compensative production of Klotho in the cardiomyocytes under stress-related conditions can be protective in cardiac tissue during ischaemic damage. Supplementation with Klotho led to an increased viability and metabolic activity of injured cells [16]. Importantly, we recently showed that Klotho protein participates in the regulation of the redox balance and supports metabolic homeostasis of the cardiomyocytes [17]. Therefore, Klotho protein may protect the cardiac cells by the inhibition of oxidative stress and subsequent degradation of contractile proteins.

Taking into account the antioxidative activity of Klotho protein [12,15], we hypothesized that Klotho contributes to the inhibition of MMPs activity and the release of MLC that follows. This study aimed to examine the influence of Klotho protein on the MMP-dependent degradation of contractile proteins in the cardiomyocytes subjected to an IRI.

## 2. Results 

### 2.1. The Level of Injury to Cardiac Cells

Lactate dehydrogenase (LDH) activity in cell supernatants (S1) was significantly higher in the IRI group compared to the aerobic control. Administration of exogenous recombinant human Klotho protein during IRI significantly reduced damage to the cardiomyocytes (Figure 1).

### 2.2. Expression of MMP Genes in Human Myocytes

The expression of the MMP-2 gene (Figure 2A) and the MMP-9 gene (Figure 2B) was significantly decreased in the myocytes from the IRI group in comparison to the cells maintained in aerobic conditions. The expression of the MMP-2 gene was over 50-fold higher than the MMP-9 gene. Incubation of the cells with Klotho during IRI regulated the expression of MMP-2 and MMP-9 genes (Figure 2A,B) to the level observed in the aerobic control group. 

### 2.3. Expression of the MMP Proteins in Human Myocytes

The expression of the MMP-2 (Figure 3A,C) and MMP-9 (Figure 3B,C) proteins was significantly increased in the myocytes from the IRI group in comparison to the cells maintained in aerobic conditions. Supplementation of the cardiomyocytes with Klotho protein during IRI slightly reduced the expression of MMP-2 and MMP-9 proteins (Figure 3A–C). 

The MMP-2 level in the cardiomyocytes was lower (Figure 4A) and the activity of 72 kDa MMP-2 protein in cell supernatants (S1) was significantly higher (Figure 4B) in the IRI group compared to aerobic control. No activity of MMP-9 protein in human cardiomyocytes was detected. The activity of MMP-2 was positively correlated with LDH activity (*p* < 0.0001, r = 0.66) (Figure 4C). There was also a negative correlation between the activity of MMP-2 protein and the expression of the MMP-2 gene (*p* = 0.0369, r = −0.54) (Figure 4D), which may suggest feedback inhibition of the MMP-2 gene expression. Incubation of cardiomyocytes with Klotho protein during IRI regulated the MMP-2 protein level (Figure 4A) and significantly reduced its activity in the extracellular space (Figure 4B).

### 2.4. Release of Contractile Proteins from Cardiac Myocytes

Incubation of the cardiomyocytes with Klotho protein during IRI significantly reduced the amount of ventricular isoform of myosin light chain 1 protein (MLC1, also known as MYL3) released (Figure 5A). The concentration of MLC1 in cell supernatants (S1) was positively correlated with MMP-2 activity (*p* = 0.0044, r = 0.64) (Figure 5B).

Klotho protein significantly influenced the degradation and release of troponin I (TnI) from the cells during IRI (Figure 6A). There was a moderate positive correlation between the TnI release and MMP-2 activation (*p* = 0.0212, r = 0.41) (Figure 6B).

## 3. Discussion

It has been well observed that one of the main factors contributing to the pathogenesis of heart IRI is the generation of ROS and RNS, the overproduction of ONOO^−^, the activation of MMPs, and then the degradation of the heart contractile proteins [1,3,4,18]. Based on recent research, Klotho protein is recognized as having a renoprotective effect and is used as a biomarker of kidney injury [13,19,20]. There is some evidence of Klotho deficiency and the occurrence of kidney disorders in animal models and in humans [19,21]. Importantly, cells that overexpressed Klotho were more resistant to oxidative stress and damage. Klotho protein had a distinct capacity to inhibit the generation of ROS, and to enhance the production of superoxide dismutase—a superoxide neutralizer that detoxifies ROS [12,14,15,22]. The protective effect of Klotho administration or overexpression in ischaemic injury to the brain and the kidneys was also shown [19,20,23,24]. In our previous research, we showed that Klotho improved the viability and metabolic activity of the cardiomyocytes during IRI, and was proposed as a biomarker of heart damage [16]. We then showed a supported redox balance by means of Klotho protein in injured cardiac cells [17]. For this reason, we hypothesized that through the reduction of oxidative stress, Klotho may break the cascade involved in IRI. The main aim of this study was to show the influence of Klotho protein on MMP activity and the level of contractile proteins in the cardiomyocytes during an IRI. Our research revealed that Klotho protein participated in a decrease in MMP-2 activation and hence in MLC1 degradation. Therefore, we suggest that administration of Klotho protein prevents the cardiomyocytes from ischaemia/reperfusion injury.

MMPs are the proteolytic enzymes that degrade the extracellular proteins and play an important role in remodeling the extracellular matrix [6]. MMPs participate in a variety of physiological processes including morphogenesis, cartilage and bone repair, wound healing, cell migration, and angiogenesis [6]. The well-known MMPs are MMP-2 (gelatinase A) and MMP-9 (gelatinase B). MMPs are synthesized as pre-proenzymes, secreted from the cell as proMMPs, and activated by the plasma or opportunistic bacterial proteinases [6]. However, in the state of oxidative stress, ROS and ONOO^−^ can react with the cysteine of the cysteine switch in the proMMP and activate it without proteolytic removal of the propeptide domain [5,25,26]. Thus, oxidative stress induces activation of MMPs, and mediates the degradation of contractile proteins, microvascular damage or myocardial injury [5,18,27]. Recent evidence shows that MMP-2 is localized within the sarcomere in the cardiomyocytes [5]. Research with animal models of cardiac ischaemia has demonstrated increased activation of MMPs and their role in the damage that occurs during myocardial infarction. In the hearts subjected to IRI, MMP-2 and MMP-9 degraded such intracellular contractile proteins as troponins, myosin light chains and titin [5,27,28,29,30,31]. MLC1 can be nitrated, S-nitrosylated (by ONOO^−^), as well as phosphorylated, which increases its affinity to the proteolytic enzymes [7,30,31,32,33]. It was found that ischaemia-induced peroxynitrite-dependent nitration/nitrosylation of MLC1 enhanced its degradation by MMP-2. The degradation of MLC1 led to heart contractile dysfunctions [4,7,18,31,33]. 

We have previously reported that inhibition of the MMP-2 activity, as well as MLC phosphorylation and nitration/nitrosylation, protected the rat cardiomyocytes against IRI [34]. Our further studies showed that the administration of the nitric oxide synthase, MMP-2 and myosin light chain kinase inhibitors led to cardioprotection in the human cardiomyocytes or the isolated rat hearts [7,35,36]. We then proved that Klotho protein participates in the regulation of a redox balance and supports metabolic homeostasis of the cardiomyocytes [17]. In the current research, increased injury in the cells subjected to IRI was shown. The cell surface expression of MMP-2 and MMP-9 proteins, and the activity of MMP-2 protein in the extracellular space were increased in the cardiomyocytes from an IRI group. The level of MMP-2 protein was lower in the IRI group and MMP-2 activity correlated with the level of cell injury, which shows the release of MMP-2 from the cells and its activation. MMP-2 is secreted from the cells as latent zymogens and then is processed by the combined action to an enzymatically active form in the extracellular space [5,37]. It is known that IRI leads to cell necrosis which disrupts the plasma membrane and releases cellular components [38]. We suggest that during IRI, MMP-2 proteins were secreted into supernatants due to damage and loss of membrane integrity. It explains the lower level of MMP-2 in the cardiomyocytes and its activation in the extracellular space. Similarly, the acute release of MMP-2 during reperfusion after ischemia contributed to cardiac mechanical dysfunction in the rat heart. MMP-2 proteins were released into the coronary effluent [39]. Reperfusion following cardioplegia activated MMPs in the plasma of patients undergoing coronary artery bypass grafting [40]. In our research, the expression of MMP-2 and MMP-9 genes was reduced in the cardiomyocytes subjected to IRI. There was also a negative correlation between the MMP-2 activity and the MMP-2 gene expression. We hypothesize that these discordant protein and mRNA levels may be due to the negative feedback on the mRNA transcription or the presence of other regulatory mechanisms that are not fully understood currently. We also speculate that the MMP-2 protein may become accumulated over time, while the MMP-2 mRNA turnover may be rapid under IRI conditions [41,42]. Our observations are supported by previous results from Chen et al. [43]. They noticed a subset of proteins in the lung adenocarcinomas that demonstrated a negative correlation with the mRNA expression values. Furthermore, the transcription and translation of proteins can be differently regulated [43,44]. The proteins may differ substantially in their in vivo half-lives due to post-translational modifications such as phosphorylation, acetylation or glycosylation, and then a reduced rate of degradation [44,45]. Importantly, it was reported that microRNAs (miRNAs) can destabilize the mRNA or stimulate rather than inhibit translation [46]. There are also RNA-binding proteins that control the translation efficiency of mRNAs [47]. The specialized mechanisms that allow certain mRNAs encoding transcription factors to be upregulated translationally under stress conditions were shown as well [46]. Another hypothesis of negative correlation between the MMP-2 activity and MMP-2 gene expression may be the constitutive production of MMP-2. Continuous production of MMP-2, regardless of the needs of cells, was observed in a wide range of cell types including cardiomyocytes, endothelial cells, macrophages, mesenchymal stem cells, and many malignant cells [37,48,49]. Constitutively produced MMP-2 could be partially activated due to cell stress. However, lower expression of the MMP-2 gene during IRI observed in this study requires further investigation. Supplementation of the cardiomyocytes with Klotho during IRI reduced cell damage, regulated the expression and level of MMP-2, and reduced MMP-2 activation, showing cardioprotection. Importantly, data provided evidence of a relationship between Klotho and MMPs. It was disclosed that a *kl* gene deficiency has been linked to the increased arterial, pulmonary and renal expression of MMP-2 and MMP-9 in animal models [50,51,52]. Induced expression of the *kl* gene led to a decrease in MMPs level in the kidneys or cervical cancer cells [50,53]. Similarly, administration of Klotho protein resulted in a lower renal level of MMP-2 and -9 [54,55]. Taking into account the above, we suggest that Klotho protein may protect the heart during IRI by indirect contribution to the MMPs expression/activation.

During oxidative stress, the level of phosphorylation, nitration, and nitrosylation of the myocardial contractile proteins is increased. Phosphorylated and nitrated/nitrosylated MLCs are then degraded by MMPs, which leads to cardiac contractility dysfunctions [7,30,31,32,36]. The intensity of MLC1 degradation was associated with the level of mechanical dysfunction in the ischaemic heart [31]. We have previously shown that the inhibition of MLCs posttranslational modifications, as well as the reduction of MMP-2 activity, provided cardioprotection from IRI in the isolated rat heart [7,35,36,56]. In the present study, the level of MLC1 in the extracellular space positively correlated with MMP-2 activity, suggesting degradation of MLC1 by MMP-2 during an IRI. There was also a moderate positive correlation between TnI release and MMP-2 activation. Importantly, degradation and release of MLC1 and TnI were reduced in the cardiomyocytes supplemented with Klotho protein during IRI. Taking into account the antioxidative activity of Klotho, it may prevent heart contractile disorders by the limitation of oxidative stress and subsequent limiting of MMP-dependent degradation of MLCs. 

## 4. Materials and Methods

### 4.1. Cell Culture

The primary Human Cardiac Myocytes (HCM) from ScienCell Research Laboratories (Carlsbad, CA, USA) were grown at 37 °C in a water-saturated, 5% CO_2_ atmosphere in Dulbecco’s Modified Eagle’s Medium (Sigma-Aldrich, St. Louis, MO, USA) containing Cardiac Myocyte Growth Supplement (ScienCell Research Laboratories, Carlsbad, CA, USA), 5% foetal bovine serum, 100 U/mL penicillin, 100 μg/mL streptomycin (all from Sigma-Aldrich, St. Louis, MO, USA). Cells were passaged at 90% confluence by harvesting with 0.25% trypsin-EDTA (Sigma-Aldrich, St. Louis, MO, USA).

### 4.2. In Vitro Chemical Ischaemia/Reperfusion Injury to Human Cardiomyocytes

Human cardiomyocytes underwent in vitro chemical IRI [57] according to the experimental protocol shown in Figure 7. The HCM underwent 15 min of oxygenation, 15 min of in vitro chemical ischaemia, and 20 min of reperfusion [7,16], in the absence or presence of 1 µg/mL [14,16] of recombinant human Klotho protein (R&D Systems, 5334-KL-025). 4-(2-hydroxyethyl)-1-piperazineethanesulfonic acid (HEPES) buffer (5.5 mmol/L HEPES, 63.7 mmol/L CaCl_2_, 5 mmol/L KCl, 2.1 mmol/L MgCl_2_, 5.5 mmol/L glucose, 10 mmol/L taurine) containing an additional 55 μmol/L CaCl_2_ and 0.75 mg/mL BSA was used for the aerobic stabilization and reperfusion of the cells. The cells were incubated in a HEPES buffer containing 4.4 mmol/L 2-deoxyglucose (to inhibit glycolysis) and 4.0 mmol/L sodium cyanide (cellular respiration inhibitor) [7,16] in the in vitro chemical ischaemia groups. The optimal ischaemia duration (15 min) was assessed by the measurement of the activity of LDH released from the cells as a marker of cell injury. In the IRI group, the HCM underwent 15 min of aerobic stabilization in the HEPES buffer at room temperature (RT). The buffer was then removed by centrifugation (1 min at 1500× *g*) and the supernatant (S1) was stored for further analysis. The cell pellet was resuspended in the ischaemia buffer and incubated for 15 min at RT afterwards. The cells were centrifuged for 1 min at 1500× *g*, the buffer was removed, and the HEPES buffer was added for 20 min at RT (reperfusion). The buffer was then removed by centrifugation at 1500× *g* for 5 min, and the cell pellet was homogenized. In the IRI + Klotho group, cells underwent IRI in the presence of Klotho protein in the buffers (1 µg/mL final concentration) during the entire IRI procedure. The cells from the aerobic control group were maintained in aerobic conditions by incubation in the HEPES buffer at RT for 50 min. 

### 4.3. LDH Activity Measurement

A lactate dehydrogenase activity assay kit (Sigma-Aldrich, St. Louis, MO, USA) was used to determine the activity of LDH in cells, according to the manufacturer’s instructions. LDH is a stable cytosolic enzyme released upon membrane damage/permeability or cell lysis and is a marker of cell damage. LDH activity was assessed in cell supernatants (S1) and normalized to total protein concentration.

### 4.4. MMP-2 and MMP-9 mRNA Expression

TRIZOL reagent (Thermo Fisher Scientific, Waltham, MA, USA) was used for the isolation of the total RNA from the HCM according to the manufacturer’s instructions. The concentration and purity of the RNA were evaluated with a microvolume ultraviolet (UV) spectrophotometer (NanoDrop Lite, Thermo Scientific). The reverse transcription of the pure RNA samples (100 ng) with iScript cDNA Synthesis Kit (BioRad, Hercules, CA, USA) was conducted to prepare cDNA according to the instructions provided. Briefly, the reverse transcription was carried out at 42 °C for 30 min, and inactivated at 85 °C for 5 min. The analysis of MMP-2 and MMP-9 genes expression was assessed with the real-time quantitative PCR (qPCR) and CFX96 Real-Time System (BioRad). The expression of glucose-6-phosphate dehydrogenase (G6PD) gene was used as an internal reference. The reaction mix (30 μL final volume) consisted of iTag Universal Sybr Green Supermix with ROX (BioRad), forward and reverse primers (250 nmol/L final conc.), water and cDNA (100 ng). The primers were designed by us and synthesized by TIB Molbiol (TIB Molbiol, Berlin, Germany). The sequences of primers 5′-3′ are as follows: MMP2F: ATCCAGACTTCCTCAGGCGG, MMP2R: CCTGGCAATCCCTTTGTATGTT; MMP9F: TTGACAGCGACAAGAAGTGG, MMP9R: CCCTCAGTGAAGCGGTACAT. 2^−ΔCt^ for calculation of the amount of particular mRNAs relative to G6PD was used. 2^−ΔCt^ was equal to the relative transcriptional mRNA level of the gene in cells that were exposed to aerobic conditions and cells subjected to IRI.

### 4.5. Immunofluorescence Staining for MMP-2 and MMP-9 Proteins

An indirect immunofluorescence staining of HCM was used to determine the expression of MMP-2 and MMP-9 proteins. Cells were cultured in a 24-well cell culture plate (Greiner Bio-One GmbH, Frickenhausen, Germany) at a density of 1 × 10^5^ cells/well. When the cell confluence reached approximately 90%, myocytes underwent in vitro chemical IRI injury in the cell culture plate (according to the protocol shown in Figure 1). Briefly, the cell culture medium was removed and cells were washed with phosphate buffered saline (PBS). Cells were then subjected to fixation at RT for 15 min with 500 μL/well of 4% paraformaldehyde. After three cycles of PBS rinsing, myocytes were incubated with the blocking buffer (1% BSA, 10% goat serum, 3M glycine in 0.1% Tween-PBS) for 1h at RT. Primary antibodies mouse anti-MMP2 1:300 (Abcam, Cambridge, United Kingdom, ab86607) and rabbit anti-MMP9 1:500 (Abcam, ab76003) were incubated with HCM at 4 °C overnight. Cells were then washed with PBS and the secondary antibodies goat anti-mouse IgG 1:500 (Abcam, ab96872) labeled with DyLight^®^ 550 and donkey anti-rabbit IgG 1:500 (Abcam, ab98488) labeled with DyLight^®^ 488 were added at RT for 45 min. To visualize cells’ nuclei, myocytes were stained with DAPI (4′,6-diamidino-2-phenylindole, Sigma-Aldrich) 1:1000 for 15 min in the dark and rinsed with PBS. A multifunction reader Spark TK Biotech 2017 (Tecan Austria GmbH, Grödig, Austria) was used to read the signal emitted. A Thunder Leica Imager (Leica Microsystems) to visualize the expression of MMP-2 (red fluorescence) and MMP-9 (green fluorescence) was used. The fluorescence of cells’ nuclei stained by DAPI (blue fluorescence) was expressed as the number of cells. The expression of MMP-2 and MMP-9 was assessed by the measurement of the fluorescence intensity of fixed cells stained with appropriate antibodies (red/green fluorescence) expressed in AU, and normalized to the number of cells (blue fluorescence).

### 4.6. Analysis of MMP-2 Concentration

The MMP-2 level in cell homogenates was assessed using quantitative Quantikine ELISA assay for total MMP-2 (R&D Systems, Minneapolis, MN, USA) according to the manufacturer’s instructions. A monoclonal antibody specific for human total MMP-2 was pre-coated onto a microplate. MMP-2 immobilized with a monoclonal antibody specific to this protein was detected with the use of an anti-total-MMP-2 polyclonal antibody conjugated to streptavidin-horse radish peroxidase (HRP). Next, 3,3′,5,5′-Tetramethylbenzidine (TMB) substrate solution was added and color developed in proportion to the amount of total MMP-2 bound. MMP-2 concentration in cell homogenates was expressed as ng per µg of total protein.

### 4.7. Zymography

Gelatine zymography for the analysis of MMP activity was performed using the protocol created in our laboratory [7]. Briefly, an aliquot of 20 μg of the total protein from the cell supernatants (S1) was mixed with the sample loading buffer (0.5 mol/L Tris-HCl/0.4% SDS pH 6.8 70% (*v*/*v*), glycerol 30% (*w*/*v*), SDS 10% (*w*/*v*), bromophenol blue 0.012% (*w*/*v*), ddH_2_O up to 10 mL) and applied to a 8% polyacrylamide gel co-polymerized with 2 mg/mL of gelatine for electrophoresis. Gels were rinsed in 2.5% Triton X-100 to remove SDS after electrophoresis. Gels were then placed in the incubation buffer (50 mmol/L Tris-HCl pH 7.4, 5 mmol/L CaCl_2_, 150 mmol/L NaCl, 0.05% NaN_3_) at 37 °C overnight for digestion. Gels were stained in the staining solution (2% Coomassie brilliant blue G, 25% methanol, 10% acetic acid) for 2 h and destained (2× for 30 min each) in the destaining solution (2% methanol, 4% acetic acid). VersaDoc 5000 (BioRad) for scanning the zymograms and Quantity One software v. 4.6.6 (BioRad) for analysis of the band intensities were used. MMP activity was expressed as the activity (AU) per µg of protein from the cell supernatant (S1).

### 4.8. The Measurement of Contractile Proteins Release

The concentration of MLC1 and TnI in the cell supernatants (S1) was determined quantitatively using the human myosin light chain 3 ELISA kit and human troponin Ⅰ ELISA kit (both from Bioassay Technology Laboratory, Shanghai, China), according to manufacturer’s instructions. Briefly, MLC1 or TnI protein from the sample was immobilized with primary human MLC1 or TnI antibody, and detected using biotinylated human MLC1 or TnI antibody. HRP and TMB substrate solution were then added, and the color developed in proportion to the amount of human MLC1 or TnI. The concentration of MLC1 or TnI in the cell supernatants (S1) was normalized to the total protein concentration.

### 4.9. Determining the Total Protein Concentration

The Bradford method [58] was used to measure the concentration of protein in the cell supernatants. BSA (heat shock fraction, ≥98%, Sigma-Aldrich) served as the protein standard. The total protein concentration was assessed using the bio-rad protein assay dye reagent (BioRad) and Spark multimode microplate reader (Tecan Trading AG, Männedorf, Switzerland).

### 4.10. Statistical Analysis

The results were analyzed with GraphPad Prism 6 software (GraphPad Software, San Diego, CA, USA). To assess the normality of variance changes, the Shapiro–Wilk normality test was used, and in all the variables *p* < 0.05 was considered significant. The comparison of data between groups was made with ANOVA or nonparametric tests with the post hoc tests. The correlation analysis was assessed with Pearson’s or Spearman’s tests. Results were expressed as mean ± SD or box-and-whisker plots, with a value of *p* < 0.05 being regarded as statistically significant.

## 5. Conclusions

We report that Klotho protein contributes to the inhibition of MMP-dependent degradation of contractile proteins and protects the cardiomyocytes subjected to IRI. We strongly suggest that Klotho protein may act as a cardiopreventive/cardioprotective and support the contractility of the cardiomyocytes during ischaemia/reperfusion.

### Limitations

We assume that our study presents a trend, but the lack of mechanisms detected precludes a precise estimation of this phenomenon. This issue may be treated as a study limitation, and therefore further research seems necessary to reveal the precise molecular mechanism by which Klotho could be involved in MMPs activity. The limitation of this work is also the use of the commercially available primary human cardiac myocytes, which may include incomplete differentiation, adaptation to the culture environment, altered cell morphology, membrane currents and contraction, changes in the expression of ion channels and contractile proteins, and reorganization of the cytoskeletal components of the cells. We are also aware that the use of cultured myocytes does not fully reflect the conditions observed in the freshly isolated myocytes or fully differentiated adult myocardium. However, while myocytes in culture are quiescent or round-shaped, the lack of classical cardiomyocyte structures did not significantly affect our research. HCM are qualified for in vitro research on cardiac diseases and for pharmacological studies, and are used routinely.

## Figures and Tables

**Figure 1 ijms-23-15450-f001:**
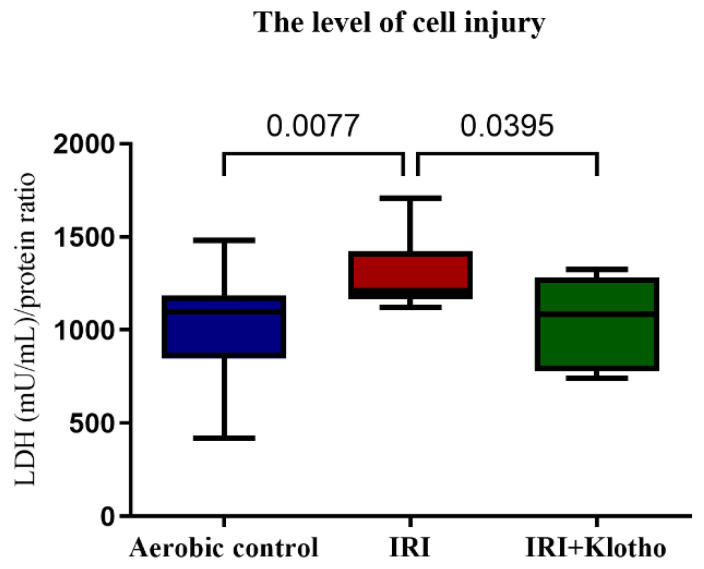
LDH activity in cell supernatants (S1) as a marker of cell death. LDH activity was normalised to µg of total protein; n_aerobic_ = 17; n_IRI_ = 17; n_IRI+Klotho_ = 15; LDH—lactate dehydrogenase; mU/mL—milli international enzyme units per milliliter; boxes—25–75% percentile, whiskers—min to max + median.

**Figure 2 ijms-23-15450-f002:**
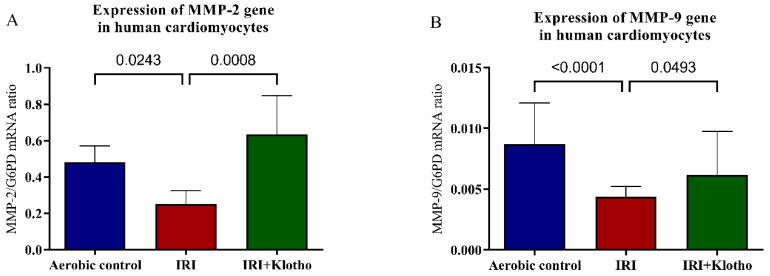
(**A**) Expression of the MMP-2 gene in human cardiomyocytes; n_aerobic_ = 5; n_IRI_ = 5; n_IRI+Klotho_ = 5 (**B**) Expression of the MMP-9 gene in human cardiomyocytes; n_aerobic_ = 14; n_IRI_ = 14; n_IRI+Klotho_ = 10. MMPs mRNA expression was examined by qPCR and normalized to G6PD expression; G6PD—glucose-6-phosphate dehydrogenase; MMP—matrix metalloproteinase; mean ± SD.

**Figure 3 ijms-23-15450-f003:**
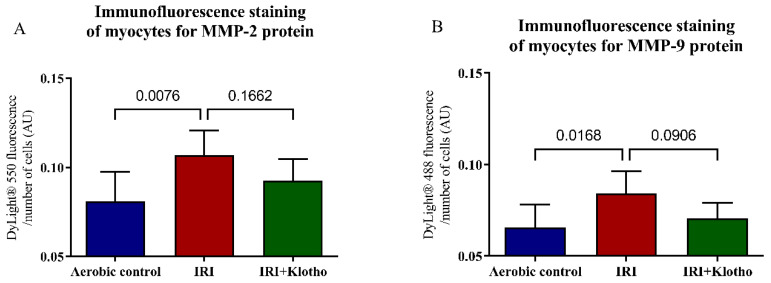
(**A**) Expression of the MMP-2 protein in human cardiomyocytes; n_aerobic_ = 7; n_IRI_ = 7; n_IRI+Klotho_ = 7; (**B**) Expression of the MMP-9 protein in human cardiomyocytes; n_aerobic_ = 7; n_IRI_ = 7; n_IRI+Klotho_ = 7. The expression of the MMP proteins was assessed by immunofluorescence staining, expressed as AU, and normalized to the total number of cells (blue fluorescence). Graph bars show the average of total fluorescence of cells in each experiment; (**C**) Immunofluorescence staining of human cardiomyocytes for MMP-2 (red fluorescence), MMP-9 (green fluorescence) and DAPI for nuclei (blue fluorescence); magnification 400× and 100×; scale bar 100 µm; AU—arbitrary unit; DAPI—4′,6-diamidino-2-phenylindole; MMP—matrix metalloproteinase; mean ± SD.

**Figure 4 ijms-23-15450-f004:**
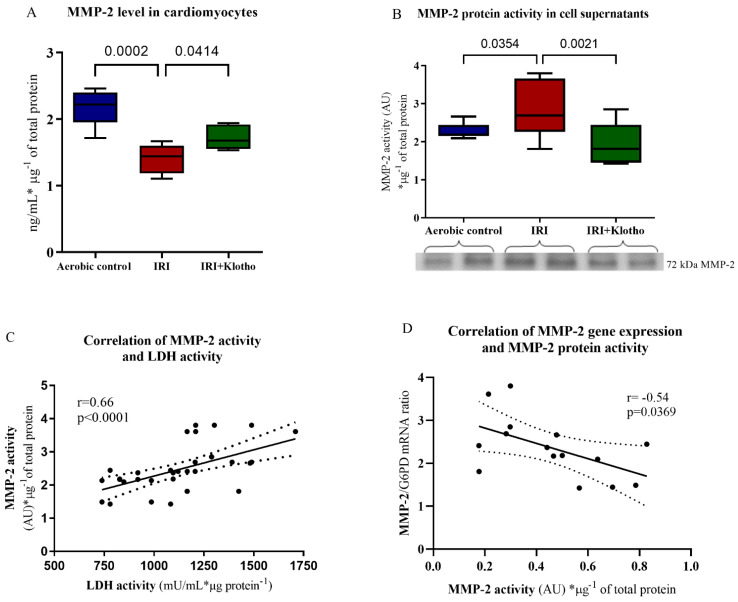
(**A**) The concentration of MMP-2 protein in cell homogenates; n_aerobic_ = 6; n_IRI_ = 6; n_IRI+Klotho_ = 5; (**B**) The activity of 72 kDa MMP-2 protein in cell supernatants (S1). The activity of MMP was examined by zymography, expressed in AU and normalized to the total protein concentration; n_aerobic_ = 10; n_IRI_ = 10; n_IRI+Klotho_ = 12; (**C**) Correlation of MMP-2 activity and LDH activity; (**D**) Correlation of MMP-2 gene expression and MMP-2 activity; lines—mean; dotted lines—standard error; AU—arbitrary unit; G6PD—glucose-6-phosphate dehydrogenase; LDH—lactate dehydrogenase; mU/mL—milli international enzyme units per milliliter; MMP—matrix metalloproteinase; boxes—25–75% percentile, whiskers—min to max + median.

**Figure 5 ijms-23-15450-f005:**
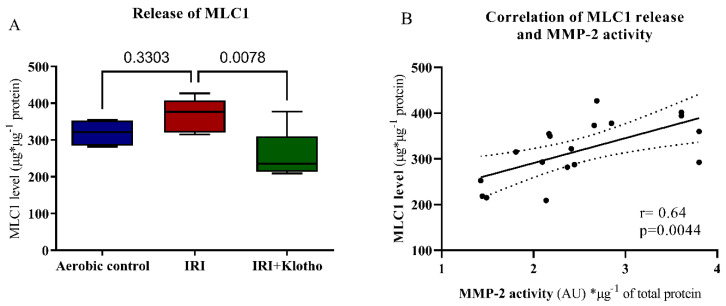
(**A**) The concentration of MLC1 in cell supernatants (S1) was tested by ELISA and normalized to the total protein concentration; n_aerobic_ = 5; n_IRI_ = 7; n_IRI+Klotho_ = 6; (**B**) Correlation of MLC1 concentration in cell supernatants (S1) and MMP-2 activity; lines—mean; dotted lines—standard error; AU—arbitrary unit; MLC1—myosin light chain 1; MMP-2—matrix metalloproteinase 2; boxes—25–75% percentile, whiskers—min to max + median.

**Figure 6 ijms-23-15450-f006:**
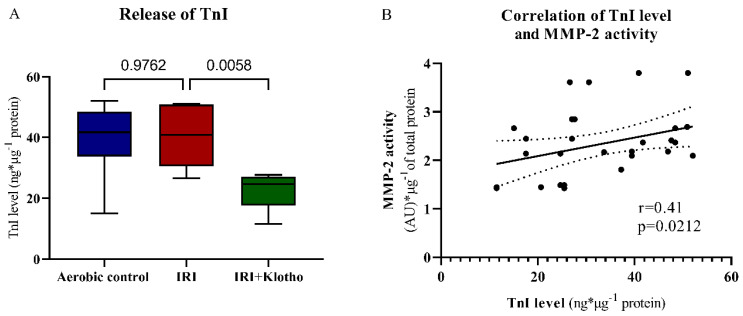
(**A**) The concentration of TnI in cell supernatants (S1) was tested by ELISA and normalized to the total protein concentration; n_aerobic_ = 10; n_IRI_ = 10; n_IRI+Klotho_ = 12; (**B**) Correlation of TnI concentration in cell supernatants (S1) and MMP-2 activity; lines—mean; dotted lines—standard error; AU—arbitrary unit; MMP-2—matrix metalloproteinase 2; TnI—troponin I; boxes—25–75% percentile, whiskers—min to max + median.

**Figure 7 ijms-23-15450-f007:**
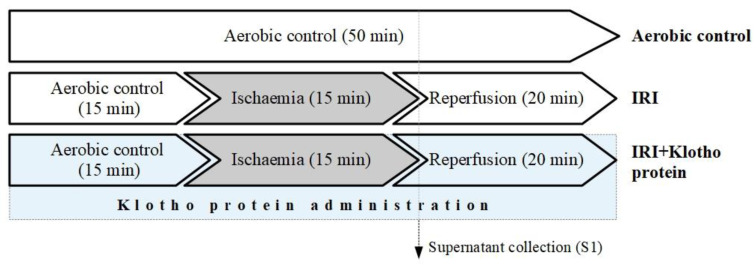
Experimental protocol for in vitro chemical ischaemia/reperfusion injury to the cardiomyocytes, with and without Klotho administration; IRI—ischaemia/reperfusion injury.

## Data Availability

The data that support the findings of this study are available from the corresponding author upon reasonable request.

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
