# Peer review of "Klotho Protein Decreases MMP-Mediated Degradation of Contractile Proteins during Ischaemia/Reperfusion Injury to the Cardiomyocytes"

_ijms, 2022, doi:10.3390/ijms232415450_

Round 1
Reviewer 1 Report
Review of the article “Klotho protein decreases MMP-mediated degradation of contractile proteins during ischaemia/reperfusion injury of the cardiomyocytes”
The study was conducted on a topical issue and is of undoubted scientific interest. Recently, the Klotho protein has received much attention in research both in pathophysiological and pharmacological aspects.
The design is logical, the experiments are performed at a high level, the goal set within the framework of the study has been achieved. However, as the authors noted, the additional studies in vivo are needed to confidently speak about the cardioprotective effect of the Klotho protein during ischemia/reperfusion.
The comments:
1. In the section “Results”, the authors wrote «Klotho protein significantly influenced on degradation and release of TnI from the 238 cells during IRI (Fig. 6A). There was a positive correlation between TnI release and 239 MMP-2 activation (p=0.0212, r=0.41) (Fig. 6B), confirming the role of MMP-2 in cardiac 240 damage during IRI».
In our opinion, such a weak correlation r=0.41 should not be taken into account. It is necessary to pay attention to the correlation between the other indicators, in all cases it does not reach 0.7. This should only be interpreted as medium-strength relationships between variables.
2. What can explain the lack of difference in TnI and MLC1 concentration between the aerobic control and IRI groups? Were they also used as markers of damage to cardiomyocytes? Probably damage was absent?
3. The clarification of the correlation analysis is needed; how was it performed? Why does the number of samples in the figure captions not match the number of points on the graphs analyzing the correlation? What does n mean and what do these dots mean?
4 Section 4 is called Discussio, it is an error.
The article needs to be improved.
Reviewer 2 Report
Limitation of the study is that it is all in vitro using cell lines and not primary cells. The cells used do not appear to have classical cardiomyocyte structures or morphology.
· Westerns should be used to assess MMP protein levels. Immunofluorescence staining did not show the changes clearly. Zymography only showed very modest differences in activity
· Effect on cell survival should be assessed
· Release of contractile proteins is associated with cellular necrosis. This show be examined.
· Was cellular proliferation altered?
· Many correlations are presented but causal relationships between them are not shown
Moderate editing for English/grammar is required (see below):
· “FGF23. It shows the 47 activity of β-glucuronidase and sialidase, and modifies the renal and intestinal ion 48 transporters12.” What is meant by this sentence is unclear. Appears to be a sentence fragment.
· “We have previously reported that compensative production of Klotho takes 51 place in the cardiomyocytes subjected to IRI17.” Not clear what is meant by compensative in this sentence
·
Round 2
Reviewer 2 Report
My previous concerns were addressed.